# JPA: Joint Metabolic Feature Extraction Increases the Depth of Chemical Coverage for LC-MS-Based Metabolomics and Exposomics

**DOI:** 10.3390/metabo12030212

**Published:** 2022-02-26

**Authors:** Jian Guo, Sam Shen, Min Liu, Chenjingyi Wang, Brian Low, Ying Chen, Yaxi Hu, Shipei Xing, Huaxu Yu, Yu Gao, Mingliang Fang, Tao Huan

**Affiliations:** 1Department of Chemistry, Faculty of Science, University of British Columbia, Vancouver Campus, 2036 Main Mall, Vancouver, BC V6T 1Z1, Canada; jian@chem.ubc.ca (J.G.); samshen0420@gmail.com (S.S.); sherry.cjy.wang@gmail.com (C.W.); brian.jlow@gmail.com (B.L.); ying1380@chem.ubc.ca (Y.C.); yaxi.hu@hc-sc.gc.ca (Y.H.); philipxsp@hotmail.com (S.X.); yhxchem@outlook.com (H.Y.); 2School of Civil and Environmental Engineering, Nanyang Technological University, Singapore 639798, Singapore; lium0025@e.ntu.edu.sg (M.L.); mlfang@ntu.edu.sg (M.F.); 3Department of Pharmaceutical Sciences, College of Pharmacy, University of Illinois at Chicago, Chicago, IL 60612, USA; yugao@uic.edu

**Keywords:** untargeted metabolomics, exposomics, feature extraction, data processing, metabolite annotation

## Abstract

Extracting metabolic features from liquid chromatography-mass spectrometry (LC-MS) data has been a long-standing bioinformatic challenge in untargeted metabolomics. Conventional feature extraction algorithms fail to recognize features with low signal intensities, poor chromatographic peak shapes, or those that do not fit the parameter settings. This problem also poses a challenge for MS-based exposome studies, as low-abundant metabolic or exposomic features cannot be automatically recognized from raw data. To address this data processing challenge, we developed an R package, JPA (short for Joint Metabolomic Data Processing and Annotation), to comprehensively extract metabolic features from raw LC-MS data. JPA performs feature extraction by combining a conventional peak picking algorithm and strategies for (1) recognizing features with bad peak shapes but that have tandem mass spectra (MS^2^) and (2) picking up features from a user-defined targeted list. The performance of JPA in global metabolomics was demonstrated using serial diluted urine samples, in which JPA was able to rescue an average of 25% of metabolic features that were missed by the conventional peak picking algorithm due to dilution. More importantly, the chromatographic peak shapes, analytical accuracy, and precision of the rescued metabolic features were all evaluated. Furthermore, owing to its sensitive feature extraction, JPA was able to achieve a limit of detection (LOD) that was up to thousands of folds lower when automatically processing metabolomics data of a serial diluted metabolite standard mixture analyzed in HILIC(−) and RP(+) modes. Finally, the performance of JPA in exposome research was validated using a mixture of 250 drugs and 255 pesticides at environmentally relevant levels. JPA detected an average of 2.3-fold more exposure compounds than conventional peak picking only.

## 1. Introduction

Liquid chromatography-mass spectrometry (LC-MS) is a high-throughput analytical platform that enables the unbiased detection and quantification of small molecules in biological samples. It has emerged as a powerful tool for metabolomics to identify key biomarkers for disease prediction and decipher the molecular basis behind biological phenomena [1,2,3]. LC-MS-based metabolomics has also become a critical analysis strategy for studying the exposome, defined as the totality of environmental exposures that drive human health and disease [4,5,6]. In particular, MS operated in data-dependent acquisition (DDA) mode offers autonomous collection of both MS^1^ and MS^2^ spectra, allowing for simultaneous quantitative comparison and metabolite annotation [7,8,9]. State-of-the-art LC-MS systems are very sensitive and can generate a large amount of metabolic information, including thousands of MS^1^ scans that contain tens of thousands of unique *m/z* values. While it is possible to manually recognize metabolic features from the raw LC-MS data, omics-scale metabolic feature extraction has to rely on feature extraction programs as manual checking is tedious and time consuming. Over the past decade, various algorithms, including *cen**tWave* [10], GridMass [11], and others [12,13], have been proposed to automatically recognize the Gaussian-shaped extracted ion chromatograms (EICs) that represent real metabolic features in LC-MS data. Unfortunately, given their diverse concentrations and chemical properties, many metabolites do not present nice Gaussian-shaped EICs and thus cannot be recognized automatically, especially those at low concentrations. For these metabolites, conventional peak picking algorithms are not efficient. As a consequence, important biological information in these low-quality metabolic features might be buried in raw LC-MS data. Thus, there is a great demand to develop novel bioinformatic solutions to recognize and extract these low-quality metabolic features in order to fully unleash the analytical power of the LC-MS platform.

To achieve comprehensive feature extraction from raw LC-MS data, our philosophy is to combine feature extraction strategies that have complementary mechanisms. Conventional algorithms recognize features with good EIC peak shapes, while features with poor EICs remain obscured. To further extract these low-quality metabolic features, we implement an extraction strategy based on their MS^2^ spectral information [14]. Furthermore, we believe that a list of targeted metabolic features can be used to guide automatic extraction in raw LC-MS data. This is because, in many metabolomics applications, there are always some general expectations. For instance, when studying the altered cell metabolism under drug/chemical treatment, the detection of key metabolites involved in energy metabolism might be of high priority 6. In these cases, targeted extraction of metabolic features can be very useful and well addresses the demands of some researchers [15].

Combining these three strategies, we developed JPA (short for Joint Metabolomics Data Processing and Annotation), an R package that offers comprehensive and streamlined metabolic feature extraction and annotation. It is important to note that JPA is a versatile program that not only directly extracts metabolic features from raw LC-MS data, but also takes the results of metabolic features generated from other data processing software (e.g., XCMS, MS-DIAL, MZmine 2) and performs further feature extraction. JPA can be adapted to both targeted and untargeted metabolomics workflows. It has been comprehensively evaluated using metabolomics data generated from chemical standards and real biological samples under different LC-MS conditions. We believe that this joint strategy can be particularly useful for LC-MS-based exposome research, in which diverse EIC peak shapes occur in the LC-MS data. JPA is freely available on GitHub (https://github.com/HuanLab/JPA (accessed date: 15 January 2022)).

## 2. Results and Discussion

### 2.1. JPA Algorithms of Feature Extraction, Alignment, and Metabolite Annotation

JPA is programmed using R in version 4.0.4. Figure 1 shows the schematic workflow of JPA. JPA accepts metabolomics data in mzData, mzML, and mzXML formats, which can be converted from vendor file formats using MS-Convert [16]. Metabolic feature extraction in JPA is composed of three functions. First, JPA performs conventional peak picking based on the Gaussian-shaped chromatographic peaks of metabolic features using the *centWave* function adapted from XCMS [10,17]. This step is termed JPA-peak picking (JPA-PP). The current version of JPA-PP only contains *centWave*, and further development is ongoing to implement more peak picking algorithms for complementary metabolic feature extraction. Alternatively, users can first generate a metabolic feature table using other data processing software, such as XCMS Online, MZmine 2, or MS-DIAL [18,19,20]. Then, JPA will recognize additional metabolic features from the LC-MS data for comprehensive feature extraction.

Second, JPA extracts metabolic features with MS^2^ spectra of unique precursor ions. This step is termed JPA-MS^2^ recognition (JPA-MR). The idea of extracting missing features with MS^2^ spectra has been previously reported by our research group 14. However, significant changes have been made in JPA-MR to improve the quantitative precision of the extracted features. The detailed algorithm is presented in Figure 2. In brief, after all MS^2^ precursors within a certain *m/z* (default: 0.01) and retention time (RT) (default: 30 s) window are extracted from the raw LC-MS data, only the precursor with the highest intensity is kept. The unique precursors are then compared against the metabolic features extracted in JPA-PP to ensure that they have not been previously extracted. Next, if the unique precursor is not at the chromatographic peak apex, JPA will automatically search for the MS^1^ scan that is at the chromatographic peak apex and replace the intensity and RT of the precursor with the values from the peak apex. The precursor with replaced intensity and RT is then identified as a putative JPA-MR feature. Finally, to confirm the putative feature as a valid JPA-MR feature, its MS intensity should be higher than the local noise level by a certain threshold (default: 3-fold). The JPA-MR feature should also be found in at least 4 consecutive MS^1^ scans. We provided an R code (thresholdEstimate.R) for users to automatically determine the local noise threshold. It is available on GitHub (https://github.com/HuanLab/JPA/blob/main/thresholdEstimate.R (accessed date: 15 January 2022)).

Third, JPA can optionally perform a targeted extraction of metabolic features from a user-provided targeted list. This step is termed JPA-targeted list (JPA-TL). In this step, the metabolic features in the targeted list are first compared with the results of JPA-PP and JPA-MR to identify the unextracted targeted features. The unextracted targeted features are then directly searched for in the raw LC-MS files and extracted as JPA-TL features. If the MS signal intensity of the extracted feature is below the local noise level by a certain threshold (default: 3-fold) or any JPA-MR feature was extracted within four consecutive MS^1^ scans, it will not be considered as a valid JPA-TL feature.

Variations of the JPA workflow can be applied to process untargeted metabolomics data. Users can use either JPA-PP or other data processing software to obtain the initial feature list, depending on their peak picking algorithm preference. The format of the feature table generated by other software has to follow the template given in the user manual (part 2.2). Users can perform JPA-TL after performing JPA-MR if they have a targeted list of metabolites with known *m/z* and RT. The detailed format of the targeted list is provided in the user manual (part 4). Users can also perform JPA-TL alone if they are only interested in extracting the metabolic features from their targeted list. Sample code is available in the “Example” folder on GitHub.

After all features are extracted from each file, isotopes, adducts, and clustering groups will be recognized and annotated using CAMERA [21]. Following that, feature alignment is performed using the XCMS grouping function, groupChromPeaks(). Retention time correction is performed using adjustRtime(), and gap filling is performed using fillChromPeaks(). In the final alignment table, the isotope and adduct percentages and grouping information are also provided. These percentages provide the likelihood of a feature to be an isotope and adduct. JPA can also be used to perform metabolite annotation. To prepare for metabolite annotation, users can directly download publicly accessible MS^2^ spectral libraries (e.g., MassBank) in .msp format or follow the provided instructions for using convertMSP.R to customize an in-house MS^2^ spectral library to fit their research purposes. The detailed MS^2^ spectral similarity algorithm in dot product is presented in Appendix A. Example outputs of each abovementioned step are also given in the user manual.

It is important to note that although JPA is primarily designed to process DDA data, it can also be used to process full-scan and data-independent acquisition (DIA) data. In these cases, features generated by JPA-MR are not available, but targeted features can still be extracted from raw LC-MS data.

### 2.2. JPA Rescues Low-Abundant Metabolic Features

To demonstrate the improved performance of feature extraction by JPA-MR, we prepared a urine metabolomics sample and serial dilutions at six concentrations (dilution factors of 1, 2, 4, 6, 8, and 10). We then ran LC-MS analyses of the serial diluted urine samples in both RP(+) and HILIC(−) modes and performed feature extraction on the raw LC-MS data using JPA-PP and JPA-MR. The extracted features from both modes were combined together for the following analyses. Given that the type of metabolic feature (i.e., how the feature was extracted) is labeled in the metabolite-intensity table (PP for JPA-peak picking, MR for JPA-MS^2^ recognition, TL for JPA-targeted list), we can directly plot the number of features extracted by each of the different strategies. As shown in Figure 3A, a significant portion (25% upon averaging all six concentrations) of the metabolic features were detected using JPA-MR, but not JPA-PP. This shows that a conventional peak picking strategy is insufficient for extracting metabolic features of diverse peak shapes. An illustrative example of a rescued feature from the standard mixture is presented in Appendix A to show the distortion of peak shape negatively impacting feature extraction by the conventional algorithm. Furthermore, the total number of metabolic features decreased by 52% ((100% (original concentration) − 48% (10-fold dilution))/100% = 52%) as the dilution factor increased towards 10-fold. This decrease makes sense as the more dilute the samples are, the lower the overall metabolic concentrations, thus making low-concentration metabolites difficult to detect by LC-MS or recognize by conventional peak picking algorithms. Interestingly, despite the decreased number of metabolic features, a substantial portion of features can be rescued by JPA using its MS^2^ recognition approach, as indicated by the grey arrows in Figure 3A. On average, around 5% of the metabolic features (the fraction of features in the grey rectangle) missed by JPA-PP are rescued by JPA-MR from the more dilute urine sample. Although the total number of features detected by both JPA-PP and -MR decreased by 52%, it is less than the decrease in features detected by JPA-PP only, which is about 59% ((82% (original concentration) − 34% (10-fold dilution))/82% = 59%).

The confidence of the rescued metabolic features is always a concern. While it is possible to manually inspect each individual feature, this process would be tedious and time consuming. To investigate the quality of rescued features, we used a recently developed artificial intelligence-based feature fidelity check software, EVA, to differentiate between features with good chromatographic peak shapes and background noise. EVA was trained using 25095 EIC plots collected from 22 LC-MS-based metabolomics projects of various sample types, LC, and MS conditions and can achieve over 90% classification accuracy 21. A fidelity rate is calculated as the number of true positive features determined in EVA divided by the total number of features. As shown in Figure 3B, the fidelity rate of the metabolic features recognized by JPA-MR is slightly higher than that of the features extracted by JPA-PP, indicating the high confidence of JPA-MR and proper choice of processing parameters for JPA-MR.

### 2.3. Confidence of the Rescued Metabolic Features

Using the same urine samples, we further compared the performance of feature extraction between JPA and conventional feature extraction strategies to demonstrate the quality of features extracted by JPA. In this study, JPA-PP was used to mimic conventional metabolic feature extraction. The comparison is aimed at recognizing the unique advantages of JPA and performing a mechanistic interpretation of the performance difference. The results of the original concentration urine dataset generated in RP(+) mode and processed by the two approaches are summarized in Figure 4. Based on the results from the original concentration urine sample shown in Figure 4A, JPA is able to detect more metabolic features than conventional peak picking algorithms due to the rescue of 40% more metabolic features by MS^2^ recognition, calculated based on the number of features in the aligned feature table.

Since the rescued metabolic features usually have non-Gaussian peak shapes, it is also important to evaluate their data quality and assure that their quantitative precision is high enough for downstream statistical analysis. Figure 4B shows that the averaged RSD% (relative standard deviation) values of all the features extracted by conventional peak picking, JPA-MR, and JPA as a whole are below 15%. The RSD% of JPA-MR is approximately 3% higher than that of JPA-PP as the low-abundant metabolic features have poor peak shapes. Even though the peak shapes of the rescued metabolic features are not always ideal, they are still sufficient for quantitative analysis.

Furthermore, the features extracted by JPA-PP only and JPA were both annotated using the same MS^2^ spectral library. The annotated metabolites were then manually inspected for endogenous metabolites associated with an HMDB or KEGG ID. Only the numbers of endogenous metabolites are summarized in Figure 4C. The numbers of identified endogenous metabolites agree very well with the numbers of metabolic features. JPA facilitates more annotated metabolites than a conventional peak picking approach alone. The metabolites uniquely annotated by JPA-MR are listed in Appendix A.

We also tested the data processing speed of JPA using a desktop computer with an Intel i9-9900k CPU @ 3.60 GHz with eight cores and 32 GB memory, Windows 10 64-bit operation system, and 10 processing threads. As we can see from Figure 4D, due to the extra time spent on JPA-MR, the overall processing time is slightly longer than the time taken for carrying out peak picking only; yet, the difference is not dramatic.

In addition, we compared JPA against other well-established software, including MZmine 2, XCMS, and MS-DIAL. The parameters used for each software have been pre-optimized. The results shown in Appendix A agree with those from Figure 4, demonstrating that JPA has the highest sensitivity in terms of feature extraction and metabolite annotation. However, it is important to note that each tool has its own uniqueness in extracting metabolic features [22].

### 2.4. Robustness

Furthermore, we tested the performance of JPA using metabolomics data generated under different data acquisition rates. In particular, cycle time poses the greatest impact on the spectral acquisition time allocated for MS^1^ and MS^2^, thus influencing the number of metabolic features and annotated metabolites. Since JPA is a robust feature extraction platform composed of multiple algorithms, it is able to rescue metabolic features and generate higher quality metabolomics results even when the parameters are not optimized. This is shown using DDA data generated with cycle times of 1, 2, and 3 s while keeping all other parameters the same. The numbers of metabolic features and annotated metabolites are plotted in Appendix A. Although the JPA-PP feature number decreases with increasing cycle time, the number of JPA-MR features increases accordingly. Therefore, JPA can consistently rescue metabolites, increase the overall annotation rate, and thus facilitate higher metabolite coverage regardless of the experimental parameters.

Moreover, we tested more tolerant parameter thresholds by using a larger mass tolerance and smaller S/N ratio (mass tolerance = 30 ppm and S/N ratio = 3) to process the same data. The results generated from using the default and adjusted parameters are plotted in Appendix A. As expected, due to less stringent processing parameters, the number of features extracted by JPA-PP increases while the number of JPA-MR features decreases slightly. However, there are still a large number of features that need to be rescued by JPA. In addition, the fidelity rate (true positive rate) of the JPA-PP features drops by 10% on average, while the JPA-MR features maintain a similar fidelity rate. Therefore, JPA has its merits regardless of the data processing parameters used.

### 2.5. JPA Offers Higher Analytical Sensitivity

In analytical chemistry, sensitivity is usually determined by the performance of the analytical instrumentation. In LC-MS-based metabolomics, metabolic feature signals are automatically determined by feature extraction software. If the software is incapable of recognizing a low-abundant feature, its analytical sensitivity is diminished. This is an important concept often overlooked by researchers as many people believe that by simply using a better LC-MS system, they can achieve better analytical sensitivity for detecting more metabolic features. As demonstrated in previous sections, the performance of conventional metabolic feature extraction algorithms in recognizing low-abundant metabolic features is limited, thus leading to lower sensitivity or higher LODs. It is important to note that the poor performance of peak picking algorithms contribute to these increased LODs, and they can be rescued if better peak picking strategies are used. From that perspective, JPA meets the demand for better feature extraction. In principle, JPA should also provide better sensitivity and lower LODs for LC-MS-based metabolomics.

To demonstrate the improved analytical sensitivity attributed to a better feature extraction strategy, we prepared serial diluted endogenous metabolite standards and analyzed them in both RP(+) and HILIC(−) modes. From the raw LC-MS data, metabolic features and their MS intensities were automatically extracted using JPA and JPA-PP only (to mimic conventional metabolic feature extraction). The number of metabolites detected by each of the three different feature extraction functions of JPA (PP, MR, and TL) is shown in Figure 5A. The generated metabolic intensity tables are provided in Appendix A for HILIC(−) mode and Appendix A for RP(+) mode. The extracted peak intensities were then used to establish calibration curves, from which LOD was determined for each metabolite standard. All the intensity values were manually inspected in the raw data to assure their accuracy. Two LODs were calculated for each metabolite, by both JPA-PP only and JPA with MS^2^ recognition and targeted list, and the results are summarized in Appendix A for HILIC(−) mode and Appendix A for RP(+) mode. To visualize the improved analytical sensitivity from using JPA, we normalized the LOD values by the highest LOD of each metabolite standard and plotted them in the circular bar plot (Figure 6). On average, the LOD values calculated using JPA are up to 3425- and 15,074-fold lower than that calculated using JPA-PP only in HILIC(−) and RP(+) modes, respectively. These results suggest that JPA consistently delivers the lowest LOD for automated data processing. It also indicates that in addition to sensitive instrumentation, the feature extraction program also plays a critical role in fully revealing the analytical performance of state-of-the-art LC-MS platforms.

### 2.6. JPA in Exposome Research

Owing to its high throughput and sensitivity, LC-MS-based metabolomics has now been widely used in exposome research to study the totality of chemical exposures and their contribution to health and disease [23,24,25]. Exposure molecules usually present in low abundance, making extraction by conventional peak picking algorithms even more difficult [26,27]. As such, JPA can be a critical bioinformatic tool to detect exposure compounds for exposome research. To demonstrate the performance of JPA in exposome research, we prepared an exposome standard mixture of 250 drugs (concentration of 1 µM) and 255 pesticide standards (concentration of 1 µg/mL). The stock mixture solution was then diluted to five concentrations with a dilution factor of 5. The exposome mixtures were analyzed in RP(+) mode. The collected raw LC-MS data were then processed by JPA-PP only and JPA. The intensity results from JPA-PP only and JPA are provided in Appendix A. The number of detected metabolites of different concentrations and software are presented in Figure 5B. Specifically, the three fractions in the column of JPA results represent different feature extraction strategies. The results demonstrate that JPA is able to rescue more (8.3-fold) exposome chemicals and achieve better performance than JPA-PP only. The results of the lower concentrations show even clearer advantages of JPA in detecting low-abundant features. The number of metabolites detected by JPA in the most concentrated standard mixture is 2.05 times more than that detected by JPA-PP, while the number of metabolites detected by JPA in the lowest concentration is 2.95 times more than that by JPA-PP. These results agree with the conclusion drawn from the analyses of urine and endogenous metabolites, that JPA can dramatically increase analytical sensitivity in exposome research.

## 3. Materials and Methods

### 3.1. Metabolomics Experiments

The performance of JPA was comprehensively evaluated using LC-MS data of endogenous metabolite standards, real biological samples, and exposure drug compounds. Representative human urine samples were first used to validate the performance of JPA. The original human urine sample was acquired from a healthy male volunteer. Serial dilution was performed to prepare urine samples of different metabolite concentrations. LC separation of the urine samples was performed using a Waters ACQUITY UPLC BEH C18 Column (130 Å, 1.7 μm, 1.0 mm × 100 mm). The detailed sample preparation and LC-MS settings can be found in Appendix A. Furthermore, to determine improved analytical sensitivity using JPA-based data processing, 134 endogenous metabolite standards were used to prepare a final stock standard mixture at 8.3 µg/mL for LC-MS analysis. Reversed phase (RP) and hydrophilic interaction (HILIC) LC separations were performed using the same Waters ACQUITY UPLC BEH C18 Column and a ZIC-pHILIC column (200 Å, 5 μm, 2.1 mm × 150 mm) (Millipore Sigma, Oakville, ON, Canada), respectively. The detailed parameter settings for RP and HILIC analyses of the standard mixtures are given in Appendix A. Both urine samples and metabolite standards were analyzed on a Bruker Impact II UHR-QqTOF (ultra-high-resolution Qq-time-of-flight) mass spectrometer coupled with an Agilent 1290 Infinity II ultrahigh-performance liquid chromatography (UHPLC) system.

The study of exposome compounds was performed using the DiscoveryProbe^TM^ FDA-approved drug library containing 1971 FDA-approved drugs purchased from APEXBIO Technology. The exposome standard mixture was prepared by mixing 250 drugs (concentration of 1 µM) and 255 pesticide standards (concentration of 1 µg/mL). Five concentration levels were prepared by diluting the above-mentioned mixture four times with a dilution factor of 5. Exposure drug compounds were analyzed using an Agilent 1200 UHPLC system coupled to an Agilent 6550-quadrupole time-of-flight (qToF) mass spectrometer (Agilent Technologies, Singapore) [28,29]. Detailed LC-MS settings can be found in Appendix A. All raw data files are publicly available in the MetaboLights repository (www.ebi.ac.uk/metabolights/MTBLS2631 (accessed date: 15 January 2022)).

### 3.2. Data Analysis

The data processing parameters used in JPA are described as follows. Peak picking parameters: ppm = 10, minimum peak width = 5 in RP(+) and 10 in HILIC(−), maximum peak width = 20 in RP(+) and 60 in HILIC(−), S/N threshold = 3, mzdiff = 0.01, integration method = 1, prefilter peaks = 3, prefilter intensity = 100, noise filter = 100. MS^2^ based feature extraction parameters: mz.tol for finding potential features = 10 ppm, mass.tol and rt.tol for confirming the features in raw data = 0.01 Da and 60 s. Targeted feature extraction parameters: mass.tol and rt.tol for confirming the target features in raw data = 0.01 Da and 30 s. Alignment parameters: bw = 5, minfrac = 0.5, mzwid = 0.015, minsamp = 1, max = 100.

Metabolite annotation for urine metabolomics was performed by searching the experimental MS^2^ spectra against MS-DIAL positive mode spectral library (version 11, downloaded from http://prime.psc.riken.jp/compms/msdial/main.html#MSP (accessed on 1 September 2021), containing 290,915 MS^2^ spectra for 13,303 unique compounds). Overlap checking between the metabolic features extracted by JPA-PP only and JPA was performed using thresholds of RT difference (default: 30 s) and accurate mass difference (default: 0.01 Da).

The limits of detection (LODs) of metabolite standards were determined by first establishing linear regression curves using the MS intensities of the serial diluted standard solutions. The intensity values used for constructing the calibration curve were manually inspected to ensure the intensity values were not mistakenly recorded from adjacent peaks. Using the intensity of the lowest concentration that was recognizable by the peak picking algorithms, the MS intensity value at 3 times the signal-to-noise (S/N) ratio was estimated and fitted into the linear regression curve to estimate the concentration LOD.

The extracted ion chromatograms (EICs) of all the metabolites and exposome standards were plotted using PlotEIC.R (available on https://github.com/HuanLab/JPA/blob/main/PlotEIC.r, accessed on 1 September 2021). Representative EICs of metabolic features from both the metabolite and exposome standards generated by JPA-PP, JPA-MR, and JPA-TL are presented in Appendix A. Feature fidelity was evaluated using EVA, a deep learning model trained with over 25,000 manually inspected EICs [30].

## 4. Conclusions

In this work, we developed JPA, a robust and versatile metabolomics data processing tool that inclusively extracts different types of small molecule features regardless of their chromatographic peak shapes. JPA is a one-of-a-kind tool that confidently extracts more metabolic or exposomic features to empower the broad metabolome coverage of LC-MS-based metabolomics. Unlike other conventional data processing software, which only annotate the extracted metabolic features by their selected peak picking algorithm, JPA is able to maximize the metabolic information for metabolite annotation by including peak picking, MS^2^ recognition, and a targeted list. We believe that this novel integrated peak picking solution will bring metabolomics research to the next level, facilitating more comprehensive biological applications. In addition, we believe that the concept of joint feature extraction should be incorporated into existing metabolic feature extraction programs for more comprehensive and sensitive metabolic feature extraction.

## Figures and Tables

**Figure 1 metabolites-12-00212-f001:**
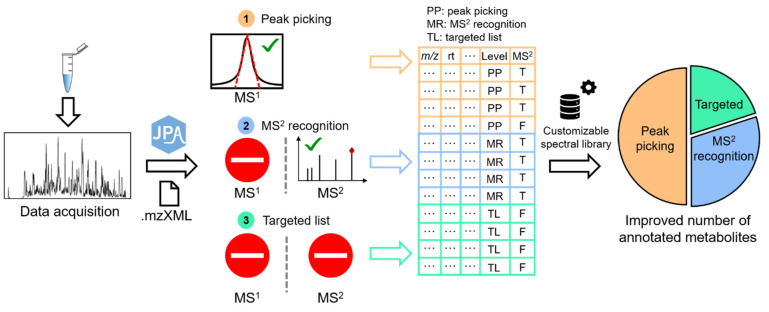
Schematic workflow of JPA (The numbers stand for the three methods of feature extraction used in JPA. The green check means the metabolic feature can be detected by using the method. The stop sign means the metabolic feature cannot be detected by using the method).

**Figure 2 metabolites-12-00212-f002:**
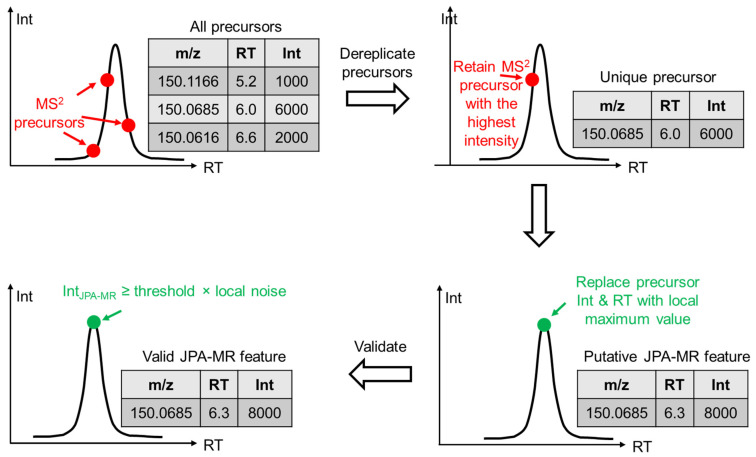
Mechanistic explanation of JPA-MS^2^ recognition (JPA-MR) in extracting metabolic features.

**Figure 3 metabolites-12-00212-f003:**
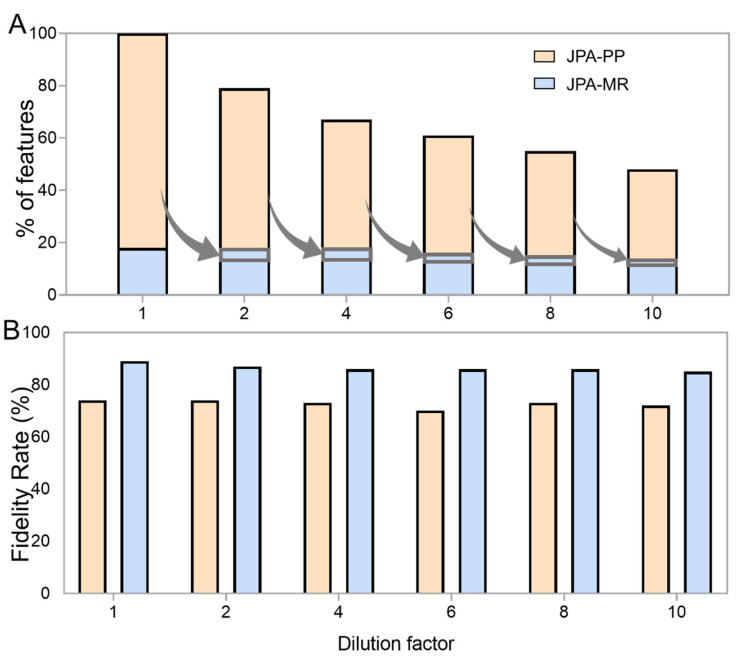
The influence of dilution on the (**A**) number and (**B**) fidelity rate of features extracted by JPA-peak picking (JPA-PP) and JPA-MS^2^ recognition (JPA-MR) in urine.

**Figure 4 metabolites-12-00212-f004:**
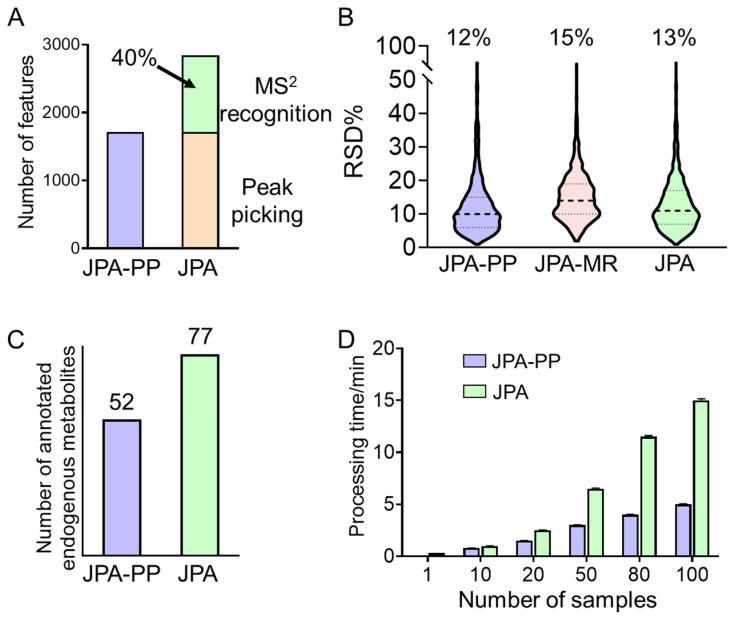
Comparison of original urine data in RP(+) mode processed by JPA-PP and JPA-MR in (**A**) number of features, (**B**) quantitative precision, (**C**) number of annotated endogenous metabolites, and (**D**) processing time.

**Figure 5 metabolites-12-00212-f005:**
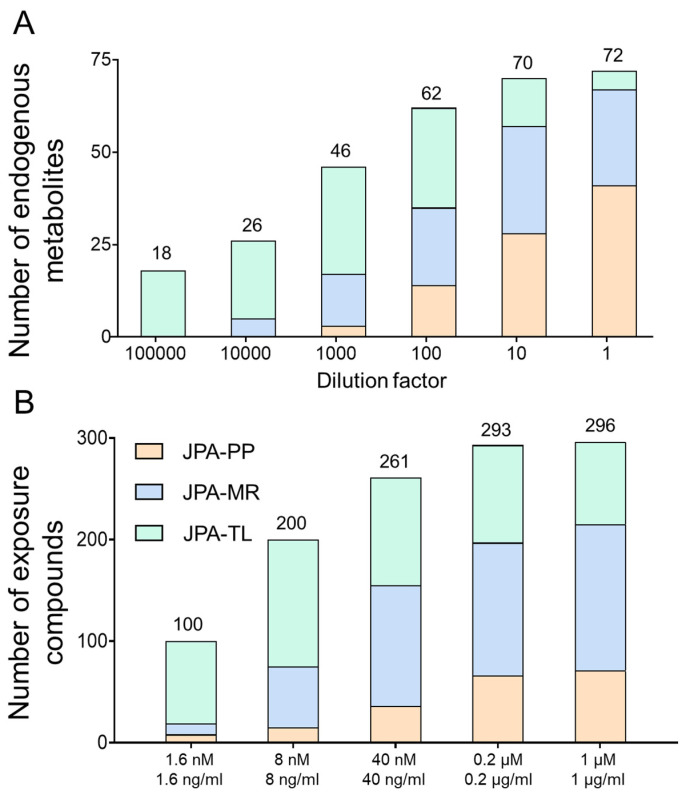
Comparison of (**A**) endogenous metabolite standards and (**B**) exposome chemicals extracted by JPA-peak picking (JPA-PP), JPA-MS^2^ recognition (JPA-MR), and JPA-targeted list (JPA-TL).

**Figure 6 metabolites-12-00212-f006:**
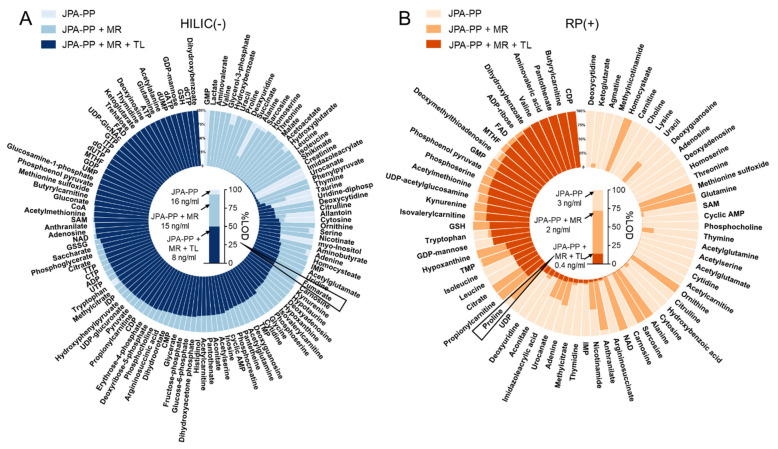
Circular bar plot of LODs of endogenous metabolites calculated based on the results of JPA-peak picking (JPA-PP), JPA-peak picking + MS^2^ recognition (JPA-PP + MR), and JPA-peak picking + MS^2^ recognition + targeted list (JPA-PP + MR + TL) on (**A**) HILIC(−) and (**B**) RP(+) mode data. The column plot inside the circular plot shows the absolute LOD of one representative metabolite in each mode.

## Data Availability

The raw data generated in this study is publicly available in www.ebi.ac.uk/metabolights/MTBLS2631, accessed on 1 September 2021.

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
