# Peer review of "JPA: Joint Metabolic Feature Extraction Increases the Depth of Chemical Coverage for LC-MS-Based Metabolomics and Exposomics"

_metabolites, 2022, doi:10.3390/metabo12030212_

Round 1
Reviewer 1 Report
The research aims to identify metabolites at low concentration that do not present Gaussian shaped EICs. This adds a new software tool that can identify these non-Gaussian shaped EICs. The paper is well written and the text is fairly easy to follow. The authors provide evidence based on their urine standards study and their validation study. They also provide a Github for their website. My comments focused on the fact that they do present a novel software that does work. However I would also take the other reviewer's comments into consideration. My focus was on the software. For me this is fine.
Reviewer 2 Report
Review of manuscript metabolites-1585395 “JPA: Joint Metabolic Feature Extraction Increases the Depth of Chemical Coverage for LC-MS-Based Metabolomics and Exposomics”
Summary
The authors present a set of tools for processing LC-MS data in this manuscript. First, the software combines different approaches for annotating compounds: exact mass, fragmentation patterns and user-defined targeted lists. Then, the authors compare the performance of the proposed tool with standard tools in the field, showing promising results. Finally, the authors present examples in the field of metabolomics and exposomics.
Commentaries
In my opinion, the research presented in this manuscript is sound, and the manuscript is well-written. The experimental data support the results and conclusions presented by the authors. Therefore, I believe that this manuscript could be eventually published in Metabolites. However, some issues should be corrected by the authors to improve the clarity of the text and the reliability of the results.
I think that the authors should clearly state if the JPA software is mostly oriented to targeted or non-targeted data analysis. The text seems that it can be used for both approaches (beginning of the results section), but the requirements for applying to one or the other should be introduced in the text.
Experimental section
- More details on the algorithm of the JPA-MR should be given. For example, how can precursors be identified in case of severe overlapping?
-What information is required on the targeted list? Please, include.
- An example output of the groupChromPeaks function could be added to the Supplementary Material.
- Give details on urine sample dilutions. How many? Dilution factor? I think that this information is only in the supplementary material.
- Include JPA processing parameters in the main text.
Results
- Section 3.1. should be moved to introduction/experimental as no results are given.
- Section 3.2. In some cases, it is not clear to me if the authors were analyzing the urine-diluted or standard-diluted samples. For instance, Figure S2: is urine or mix?
- JPA-MR. I would like to know the mass error associated with these features not identified by JPA-PP. I think that some figures of merit for the annotations should be given: mass error, identified fragments, …
- Figure S4. I think that the y-scale should be the same in both A-B / C-D graphs to show the authors' point better.
- Move Figure S-6 to the main text.
Reviewer 3 Report
The manuscript by Guo and colleagues presents an R package, named JPA, to efficiently extract metabolic features from raw LC-MS data. The method is based on the combination of conventional peak picking and recognition of bad peak shapes in MS2 spectra as well as a user-defined candidate feature. The authors demonstrated the performance of the strategy in this study and evaluated their analytical accuracy/precision. I installed the JPA package from the GitHub repo, and it did not work well. I have some suggestions and comments as follows.
- As you know, the processing LCMS-based metabolomics data is depend on their analytical platforms. The authors should describe the applicability of the software. For example, the platform names, vendors, and versions.
- I would suggest that the authors should include actual "How-to's" of the package as a vignette, GitHub website, or Supplementary materials on the journal site. It is better to include the simulated- and real data used in this paper in the package.
Round 2
Reviewer 2 Report
Review of manuscript metabolites-1585395 “JPA: Joint Metabolic Feature Extraction Increases the Depth of Chemical Coverage for LC-MS-Based Metabolomics and Exposomics” (Revision 1)
I appreciate the effort the authors made to improve the manuscript and to take into consideration the comments and suggestions.
Therefore, I consider that now the manuscript is suitable for publication. However, the authors should carefully proof-read the text to avoid typos and punctuation errors.
Author Response
We have done another round of proof reading. There should be no more typos and punctuation errors.